# Microstructure Analysis and Mechanical Properties of Backfill Material Using Stone Sludge

**DOI:** 10.3390/ma16041511

**Published:** 2023-02-11

**Authors:** Jong-Won Lee, Cheolmin Baek

**Affiliations:** Department of Highway and Transportation Research, Korea Institute of Civil Engineering and Building Technology, 283 Goyangdae-ro, Ilsanseo-gu, Goyang-si 10223, Republic of Korea

**Keywords:** stone sludge, CLSM, backfill

## Abstract

In this study, the optimum mixing ratio for backfill was determined by analyzing the mechanical properties and microstructure of controlled low-strength material (CLSM)―the backfill material for road excavation and restoration―using the total amount of stone sludge generated during aggregate production, including analysis of the flow and material separation resistance by compounding factors. A small-scale field evaluation was conducted using the derived optimal mixing ratio. A comprehensive analysis of the mechanical properties was based on the relevant standards and specifications, and a microstructure analysis indicated that the optimal mixing ratio of CLSM containing stone sludge was 350% W/B. A field applicability evaluation indicated that the flow was 230 mm, and the initial and re-excavation properties were 0.75 and 1.15 MPa, respectively, which met ASTM standards. Monitoring for approximately 5 months revealed that there was no ground subsidence caused by traffic, and it was confirmed that re-excavation with equipment was possible. After re-excavation, the filling properties of CLSM using stone sludge and the pipe periphery were checked, and it was found that no significant filling and pipe damage had occurred.

## 1. Introduction

The construction industry in metropolitan areas like Seoul, where population, traffic, and industry are concentrated, is primarily focused on maintenance and restoration of existing road pavements rather than on new construction. With the growing number of aging underground facilities, small-scale excavation and restoration works in downtown areas are continuously increasing. The government is implementing strict regulation to control the amount of fine dust and dust scatter generated during these activities. The small-scale excavation and restoration process consists of 10 stages, excluding asphalt concrete repaving. The stages that generate a large amount of dust scatter include road cutting, crushing, excavation, and soil covering [1].

As demand for construction increased in the late 1970s, the demand for concrete production and aggregates also increased significantly. In the early days of the construction industry, river aggregates were used for concrete mixing; but due to the shortage of river aggregates, the use of crushed aggregates through quarrying in mountains became inevitable. Currently, crushed aggregates account for the majority of aggregates used in concrete production. In the process of producing coarse aggregates, which are primarily crushed through quarrying, byproducts, such as stone and stone sludge, are generated. Stone dust alone accounts for approximately 30% of the raw stone. To utilize this byproduct, fine aggregates are produced from stone dust via dry or wet methods, and stone sludge is generated from approximately 30% of the stone dust [2,3].

Stone sludge generated during the production of crushed stone aggregate consists of approximately 5% raw stone, and, according to the estimates of crushed stone production, more than 15 million tons of stone sludge are generated annually. However, though stone sludge is recycled as fill and cover material, its use as a resource is currently limited. This often results in illegal landfilling or storage on business sites owing to the transportation costs and additional processing costs incurred during reclamation [4,5,6]. To address this issue, the relationship between the water–cement ratio and stone dust content was previously examined; further, the strength characteristics while using stone dust as an admixture were also studied [7]. In previous studies, it was found that incorporating stone dust in concrete can have positive effects on mechanical properties, such as strength and slump, as well as economic benefits, by reducing the amount of cement and aggregate. Lim et al. analyzed the effects of varying the mixing ratio of stone sludge on strength and flow, and observed that, as the amount of stone dust sludge added increased, the flow and strength decreased [8]. Additionally, Song et al. found that the use of fine powder (0.08 mm or less) generated during the crushing process of aggregate in concrete can lead to decreased strength and increased drying shrinkage when mixed with fine particles, such as silt and clay soil [9]. However, another study found that using less than 10% of stone sludge as a replacement for cement in concrete can result in a similar design and field application to that of existing concrete [10].

Seo et al. reviewed the usage range of fine aggregate powder, or stone sludge, with particle sizes of ≤0.08 mm, and suggested a mixing design for concrete mixed with stone sludge to increase the amount of usage [11]. Park analyzed trends in the generation of stone sludge, using data reported for the domestic aggregate supply and demand plan of each year, and further reviewed the method of calculating the appropriate water–binding material ratio of stone sludge. The results revealed that stone sludge was applied as a concrete filler to general-strength high-flowing concrete, and an expected reduction of approximately 10% in cement usage was reported. The majority of studies using stone sludge have been conducted to investigate the change in mechanical characteristics of concrete resulting from mixing and replacement of materials; and none of the cases used only stone sludge in place of cement, aggregate, and development materials for road excavation [10,12].

Ling et al. discussed the research and practical application of CLSM in backfilling. They discovered that materials used for producing CLSM vary across countries, based on a review of 115 studies from different countries. This has had a significant impact on the nature of CLSM research and its practical applications in the field [13]. As CLSM-related research has gained momentum, more studies have been conducted on the use of industrial waste materials for CLSM. For example, Zhang et al. explored the use of fly ash and coal gangue as raw materials for CLSM, and found that a mix ratio of 14:5:1 of gangue: fly ash: cement had good fluidity and met compressive strength requirements [14]. Similarly, Chen et al. used five different byproducts of the coal industry (bottom ash, fly ash, desulfurized gypsum, gasification slag, and coal gangue) and cement to prepare multi-component coal industry solid waste-based CLSM. The results showed that these mixtures satisfied the limits and requirements set by the American Concrete Institute Committee 229 (ACI) for CLSM [15,16].

The objective of this study was to enhance the utilization of stone sludge, which replaces only part of the existing concrete materials, as a sustainable and eco-friendly solution for conserving natural resources and recycling industrial byproducts. The research focused on evaluating the mechanical properties and microstructure of controlled low-strength material (CLSM), commonly used for road excavation and restoration work, by replacing a portion of the fine aggregate with stone sludge. Additionally, the study aimed to assess the field applicability of this approach through a small-scale field evaluation.

## 2. Materials and Methods

### 2.1. Materials

#### 2.1.1. Stone Sludge

The stone sludge used in this study was produced by L Company—a manufacturer of crushed aggregate in Gyeonggi-do. A total of 10 samples were tested 5 times each in the test performed to identify the characteristics of stone sludge, and the average value was used. It had a 5-mm sieve passing rate of 100%, a 0.075-mm sieve passing rate of 47.54%, and a maximum drying density of 1.694 g/cm^3^. For a characteristic analysis, engineering classification was performed according to KS F 2324 (Method for Engineering Soil Classification) [17]. According to the results, the stone sludge was classified as clay sand, as shown in Table 1. In addition, a water-content test was conducted for the utilization of the stone sludge as a backfill material, and the results are presented in Table 2.

#### 2.1.2. Cement

In this study, ordinary Portland cement manufactured by Domestic Company “Hanil” was used in compliance with the regulations of KS L 5201 [18]. Its physical properties and chemical composition are presented in Table 3.

### 2.2. Mix Proportion

In this study, to identify the mechanical characteristics of CLSM using stone sludge as a replacement for fine aggregate, the water–binder ratio (W/B = 250–500%) was determined through preliminary experimentation. The replacement rate of stone sludge was set at 100%, and the mixing conditions are outlined in Table 4. For each mixing proportion, 20 tests were conducted, resulting in a total of 120 tests. Flowability and material separation resistance were tested 20 times for each mixing proportion, and 360 samples were prepared for compressive strength testing, with 60 samples per mixing proportion.

### 2.3. Experimental Methods

#### 2.3.1. Determination of Optimal Mixing Ratio

In this study, the first step was to evaluate the use of stone sludge manufactured according to the mixing design. The flow, material separation resistance, and compressive strength were evaluated against the standards prescribed in each standard and specification. In the next step, microstructure analysis was conducted using scanning electron microscopy (SEM) and X-ray diffraction (XRD) techniques. The optimal mixing ratio was determined by evaluating the results in each step.

#### 2.3.2. Flow

In this study, the evaluation of the flowability of CLSMs was conducted in accordance with ASTM D 6103 (Standard Test Method for Flow Consistency of Controlled Low Strength Material) [19], which is the most widely used method for evaluating the flowability of CLSMs. The results of the flow test are illustrated in Figure 1.

#### 2.3.3. Material Separation Resistance

We sought to examine the “bleeding” aspect of the CLSM, as an increase in the mixing amount to ensure the flow of the CLSM using stone sludge can lead to material separation, in which a large amount of “bleeding water” is generated, depending on the density differences between the materials used. The evaluation was conducted by adopting the test method introduced in the “fluidization treatment method of soil” recommended by the Japan Fluidization Treatment Technology Management Research Association. As a detailed test method, JSCE-1986 (Japan Society of Civil Engineers) was applied [20]. After mixing the filling material, the sample was filled to a diameter of 5 cm and a length of ≥50 cm to prevent air from being mixed and was inserted into a measuring cylinder with water to determine the initial volume. Thereafter, the volume of the “bleeding water” separated into the upper part was measured, after leaving it for 3 h and 20 h, and the “bleeding” rate was calculated as a ratio to the initial sample volume.

#### 2.3.4. Compressive Strength Test

The compressive strength of the CLSM using stone sludge was measured in accordance with ASTM D 4832 (Standard Test Method for Preparation and Testing of Controlled Low Strength Material (CLSM Test Cylinders) [21]. The results of the compressive strength test are illustrated in Figure 2. A cylindrical specimen, measuring 100 mm in height and 50 mm in diameter, was prepared using a 2:1 cylinder-type mold. To evaluate the initial strength characteristics, the compressive strength at 7 days was measured, and the compressive strength at 28 days was evaluated to assess the re-excavation characteristics.

#### 2.3.5. SEM and EDS Analysis

SEM was performed to analyze the microstructure of the CLSM using stone sludge. A low-voltage long-emission scanning electron microscope (Merlin Compact, Carl Zeiss, Germany) was used, which had an in-lens detector and various signal-processing functions. In addition, energy-dispersive X-ray spectroscopy (EDS) was used to determine the elemental composition of the particle surface.

#### 2.3.6. XRD Analysis

XRD analysis was performed to analyze the substances produced in the hydration reaction of stone sludge with cement and water. XRD analysis was performed using a 1D detector (LYNXEYE) and a D8 Advance diffractometer (Bruker-AXS, Shibuya, Tokyo, Japan) equipped with a Cu target. Diffraction patterns were obtained under the conditions of 2θ = 5° to 95°, a 0.01° step size, and 1 s per step. A 0.3° divergent slit and a 2.5° secondary Soller slit were used. XRD patterns were obtained for the original specimen and the standard specimen (LaB6, SRM 660b, NIST, Gaithersburg, MD, USA) under the same conditions, to perform a qualitative analysis of the specimens and the basic parameters of the device.

#### 2.3.7. Field Applicability Evaluation

In this study, the production and pouring of the CLSM using stone sludge were conducted in the field using a forced concrete mixer, and the onsite constructability was confirmed through a characteristic analysis of the CLSM. In addition, we evaluated the initiation period of the follow-up process (asphalt concrete pavement construction) after construction, cross-sectional fillability, and quality continuity. To evaluate the initiation period of the follow-up process, a mountain-type soil hardness tester was used. After the follow-up process was conducted, the quality continuity was evaluated by monitoring it for 5 months. Subsequently, to verify the re-excavatability and fillability of the CLSM using stone sludge, re-excavation was performed, and the fillability and pipe breakage around the pipe were visually observed.

## 3. Experimental Results and Analysis

### 3.1. Flow Test Results

The quality standards for the liquidity of CLSM are stipulated in various standards and specifications in the United States and Japan. In this study, a value higher than 200 mm suggested in ACI Committee Report 229 [16] was selected as the target. For the development of CLSM using stone sludge, we attempted to verify the field applicability according to the optimal mixing ratio derived from the indoor experiment.

A liquidity test was conducted for evaluating the fillability and workability of the CLSM according to the unit binding amount of CLSM and the blending factor of W/B using stone sludge. According to the evaluation results, the liquidity tended to decrease as W/B decreased. As shown in Figure 3 all the target values were satisfied when W/B was ≥250%. For the dried soil, (S + C)/W decreased as W/B increased, and the flow decreased as (S + C)/W increased in consideration of the flow characteristics according to the ratio of (S + C)/W.

### 3.2. Material Separation Resistance Test Results

CLSM requires a large amount of water to ensure its flow characteristics, filling properties, and self-leveling. Thus, it is necessary to examine material separation resistance. The quality standard for material separation resistance of fluidized fillers was selected as a target of ≤1.0%, in accordance with Japan’s “Tokyo Construction Bureau Fluidized Soil Quality Standards (2007) [22]”. Figure 4 shows the “bleeding” test results with respect to the formulation. According to the experimental results, the “bleeding” rate was 0.76–0.84%, and there was no constant trend according to the W/B. As the W/B ratio increases, the amount of cement decreases relatively, and the amount of stone dust sludge increases. It is hypothesized that the increased clay content in the stone sludge compensated for the decrease in cement and improved the material separation resistance of CLSM. Therefore, no bleeding was observed, owing to the reduction in cement content of CLSM [8]. The quality standard for “bleeding” is <1.0%, according to the relevant standards in Japan, and all formulations in this study satisfied the W/B standard.

### 3.3. Compressive Strength Test Results

The CLSM does not require compaction, and strength is needed to prevent deformation and fracture when it is subjected to compressive and shear loads between surrounding structures and the ground. In addition, to ensure sufficient traffic opening time, the CLSM needs to secure the initiator of the subsequent process after pouring; thus, it is necessary to secure the initial strength. Currently, the compression-strength quality standard stipulates that the early compression strength of the CLSM is 0.3 MPa or higher in ACI 229R in the United States, where CLSMs are commercialized [23]. Thus, in this study, the early compression strength of the CLSM was set as ≥0.3 MPa from 7 d of age with consideration of the safety rate. In addition, we intended to secure the re-excavatability of the CLSM in case re-excavation was required. In the foreign standard ACI 229R, when aggregates with particle sizes of ≤19 mm are present in a CLSM composed of fine particles under ≤1.4 MPa, fly ash, etc., it is specified that the re-excavatability can be secured up to ≤2.1 MPa. Thus, in this study, the 28-d compressive strength was set at ≤1.4 MPa, with consideration of the characteristics of stone sludge.

According to the results for the initial strength, as illustrated in Figure 5, strength tended to decrease as the W/B increased for all combinations of the CLSM using stone sludge, and the initial strength target value was met within all applicable combination ranges. As shown in Figure 5, the compressive strength at 28 days to secure re-excavation exhibited the same trend as the initial strength, and it was confirmed that the target value was met for all formulations except for 250%. As described previously, when the flow characteristics, material separation resistance, uniaxial compressive strength, and re-excavation properties in the indoor experiment were comprehensively evaluated, it was found that the standard and target values were satisfied at W/B = 300–500%.

### 3.4. SEM and EDS Analysis Results

To analyze the microstructure according to the age of the CLSM using stone sludge, samples at 7 and 28 days were collected, and SEM was performed. The results are shown in Figure 6 and Figure 7. For the CLSM using stone sludge, as the W/B decreased and the age increased, the hydration reaction with cement became more active, and the production of ettringite, calcium silicate hydrate (C-S-H) gel, and calcium aluminate hydrate (C-A-H) gel became active [24]. For the W/B 400–500% formulation, a relatively large amount of Ca(OH)_2_ was produced at 7 days compared with the other formulations. A large amount of Ca(OH)_2_ was observed on the 28th day, and short and thick ettringite (as in the initial production state) and C-S-H gel in the form of a thin sheet with a low density were observed. Additionally, the amount of production was smaller than those for the other combinations. It is judged that a sufficient hydration reaction did not occur even after a period of time, owing to the use of little cement. For the W/B = 250–350% mixture, larger amounts of ettringite, C-S-H gel, and C-A-H gel were observed on the 7th day for W/B 400–500%, and a thin sheet-shaped C-S-H gel was widely distributed on the surface of Ca(OH)_2_. Ettringite, C-S-H gel, and C-A-H gel were actively generated even at 7 days as Ca(OH)_2_ was consumed. In particular, as the C-S-H gel and C-A-H gel were actively generated (centered on the ettringite nucleus) at 28 days, it was confirmed that the micropores of the CLSM using the stone sludge were filled, stabilizing the internal structure [25].

### 3.5. XRD Analysis Results

To analyze the microstructure according to the age of the CLSM using stone sludge, samples at 7 and 28 days were collected and subjected to XRD analysis. The results are shown in Figure 8 and Figure 9. Because stone sludge was used, the peak of SiO_2_ was clearly observed for all the formulations [26,27]. Ca(OH)_2_, ettringite, C-S-H gel, and C-A-H gel hydrates generated using cement as a binder were confirmed, and the peak intensities increased with the age of the mixture. The crystallinity of the CLSM using stone sludge was as low as 40% at approximately 7 days, but increased by 22% to 62% at 28 days, indicating that the formation of crystallites was well accomplished. In agreement with the SEM analysis, crystalline hydrates such as ettringite, Ca(OH)_2_, C-S-H gel, and C-A-H gel were actively produced from the initial stage, and the amount of production and the amount detected increased over time. 

### 3.6. Derivation of Optimal Mixing Ratio

We sought to derive the optimal mixing ratio that would satisfy the criteria stipulated in each standard and specification for CLSM using stone powder produced according to the mixing design based on the results of indoor tests, e.g., the flow, material separation resistance, and compressive strength. The flow test revealed that all the formulations satisfied the criterion of ≥200 mm suggested by ACI Committee Report 229 [16]. In the case of material separation resistance, the required performance of the backfill material stipulated in the relevant Japanese standards was <1.0%, which was found to be satisfactory in the formulation design W/B in this study. In the case of the compressive strength, it was found that the target values of ≥0.3 MPa at 3 days and ≤1.4 MPa at 28 days suggested by ACI Committee Report 229 [16] were met by all formulations except for 250%. The microstructure analysis results indicated that ettringite, C-S-H gel, and C-A-H gel―which are advantageous for securing the initial strength―were actively produced, and as C/B 250–350% of W/B was actively produced on the ettringite nucleus on the 28th day, the internal pore structure of the C-S-H gel was stabilized. Therefore, the comprehensive analysis of the mechanical properties and microstructure of the CLSM using stone sludge indicated that the optimal formulations were 300% W/B and 350% W/B. However, in the case of the W/B 300% formulation, the error range of the 28-day compressive strength exceeded the standard strength of 1.4 MPa; thus, the optimal formulation to be used for the final field evaluation was selected as W/B 350%.

### 3.7. Field Applicability Evaluation

In this study, the W/B 350% mixing ratio was applied to a field applicability evaluation performed using the optimum formulation derived from the indoor experimental evaluation, and the process and results are presented in Figure 10 and Table 5. The most commonly used concrete forced mixer in the field was used for onsite production and pouring of the CLSM using stone sludge. The flow was assessed to examine the characteristics of the CLSM using the produced stone sludge, and specimens were manufactured onsite. In addition, the change in hardness was observed using a soil surface penetrometer after the CLSM was placed. After 28 d, the stability of the CLSM was assessed through asphalt pavement, and the cross-section of the excavated part was reviewed to examine the fillability of the fluidized filler.

Regarding overseas standards for securing the initiation period of CLSM’s follow-up process, in the case of Japan (Quality Standards of Flowing Soil in Construction Bureau of Tokyo) (2007) [22], if the penetration level of the soil surface penetrometer is >3 mm, the hardness is sufficient to initiate the follow-up process As show in Table 5, the hardness characteristics were evaluated over time using a soil surface penetrometer, and it was found that the standard value (≥3 mm) was satisfied within approximately 2 h after pouring.

Figure 11 presented the comparison of test result measured in the lab and field. The result of the flow test conducted after field production was 230 mm, which was approximately 12% lower than that of the indoor test; but it was found that the target flow of ≥200 mm could be secured. Evaluating the compressive strength of the specimen manufactured during field application of the CLSM using stone sludge revealed that the initial strength was 0.75 MPa and the re-excavatability was 1.15 MPa, and the re-excavatability could be maintained below 1.4 MPa as stipulated in ACI 229R [16]. The compressive strength of CLSM in the field test showed an average reduction of about 10% when compared to the laboratory test results. The flowability and compressive strength of CLSM were found to be lower in the field test than in the laboratory test. This difference is attributed to the variations in the water content of the stone sludge caused by the field weather conditions, which were not present in the laboratory test, and to the variations in efficiency between the mixers used in the laboratory and in the field [28,29].

As shown in Figure 12, After 28 d, asphalt pavement was performed to examine the stability of the CLSM. During paving, subsidence of the ground did not occur, and examining the cross-section of the excavation site for later fillability assessment revealed that the CLSM using stone sludge was well filled around the pipe [30,31,32].

## 4. Conclusions

The mechanical characteristics and microstructure were analyzed for different formulation designs of CLSM using stone sludge, and the field applicability was assessed through a small-scale field evaluation. The results are as follows. 

(1)Examining the mechanical properties according to the mixing conditions of the CLSM using stone sludge revealed that as the W/B increased, the flow increased and the strength decreased. The flow and strength were satisfied in the range of W/B of 300–500% among the formulations applied to derive the optimal mixing ratio.(2)A microstructure analysis revealed that, as the W/B decreased and the age of the CLSM using stone sludge increased, the production of ettringite, C-S-H gel, and C-A-H gel increased. In particular, for the mixture with W/B = 250–350%, the internal structure was stabilized as the C-S-H gel was actively generated around the ettringite nucleus on the 28th day of age, and the crystalline structure was well generated; thus, the crystalline structure was confirmed to be approximately ≥60%.(3)A comprehensive analysis of the mechanical characteristics and a microstructure analysis of the CLSM using stone sludge produced according to the mixing design presented in this study revealed that the optimal mixing ratio of CLSM using stone sludge was W/B 350%.(4)The field applicability evaluation indicated that the flowability was 230 mm, and the initial and re-excavation properties were 0.75 and 1.15 MPa, respectively, which met the standard. Additionally, the hardness test standard was met after approximately 2 h of CLSM pouring, indicating that the subsequent process could be started. However, the moisture content of the stone sludge, field equipment, and weather conditions should be considered.(5)Monitoring for approximately 5 months after paving the asphalt revealed that ground subsidence due to traffic did not occur, and it was confirmed that re-excavation through equipment was possible. After re-excavation, the filling properties of the CLSM using stone sludge and the pipe periphery were checked, and it was found that significant filling and pipe damage did not occur.

## Figures and Tables

**Figure 1 materials-16-01511-f001:**
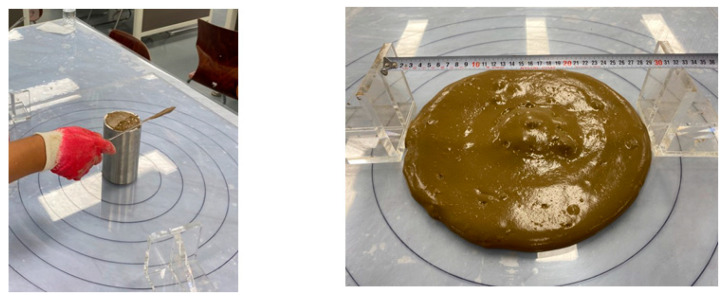
Flow test of CLSM using stone sludge.

**Figure 2 materials-16-01511-f002:**
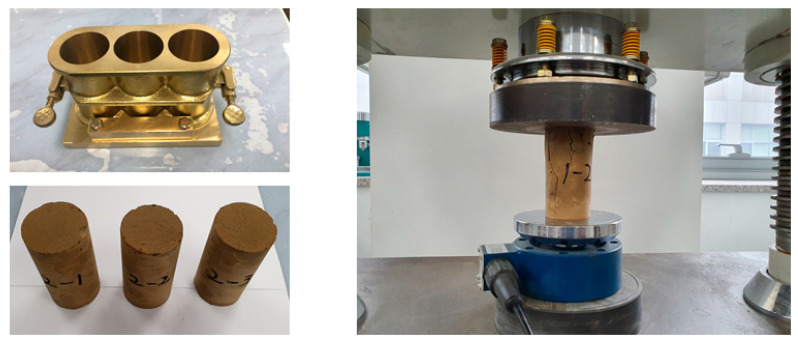
Compressive strength test of CLSM using stone sludge.

**Figure 3 materials-16-01511-f003:**
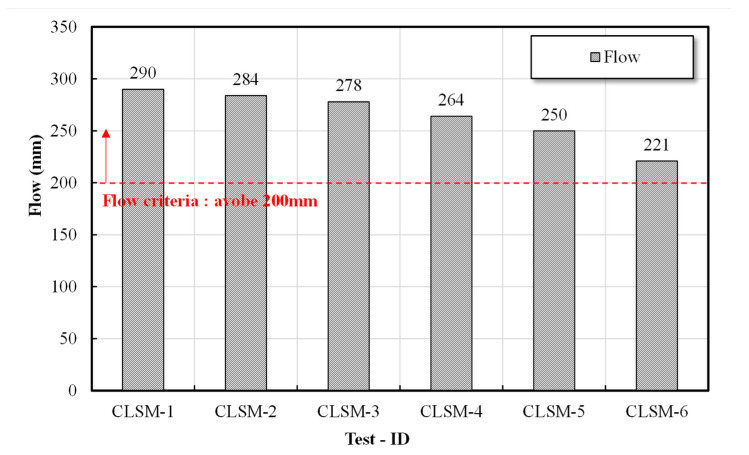
Flow test results based on the W/B.

**Figure 4 materials-16-01511-f004:**
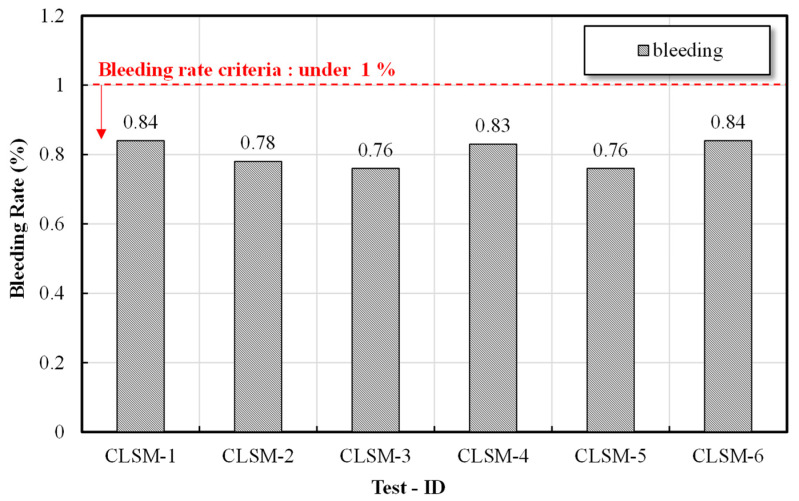
Material separation resistance test results based on the W/B.

**Figure 5 materials-16-01511-f005:**
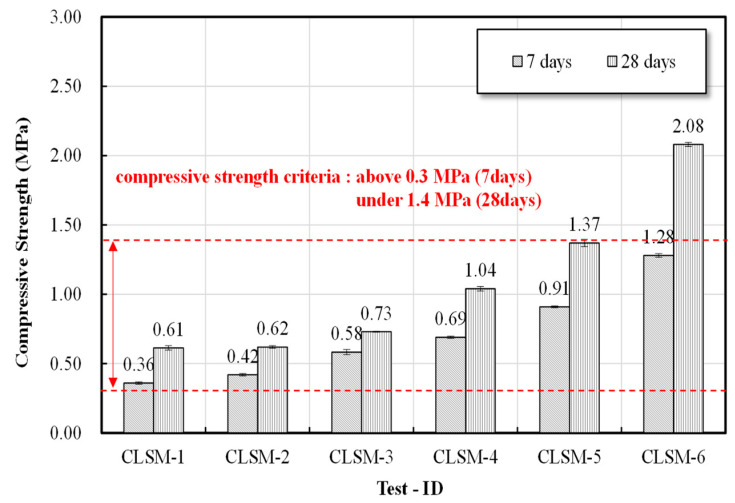
Compressive strength test results by age with respect to the W/B.

**Figure 6 materials-16-01511-f006:**
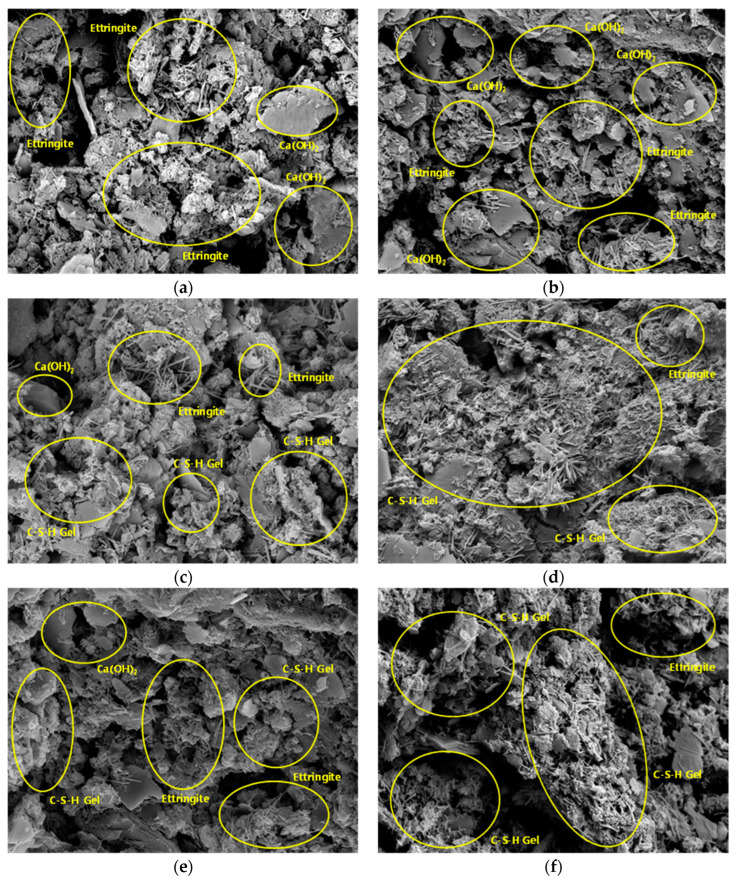
7-d SEM images for different W/B values (×10,000). (**a**) W/B = 500%; (**b**) W/B = 450%; (**c**) W/B = 400%; (**d**) W/B = 350%; (**e**) W/B = 300%; (**f**) W/B = 250%.

**Figure 7 materials-16-01511-f007:**
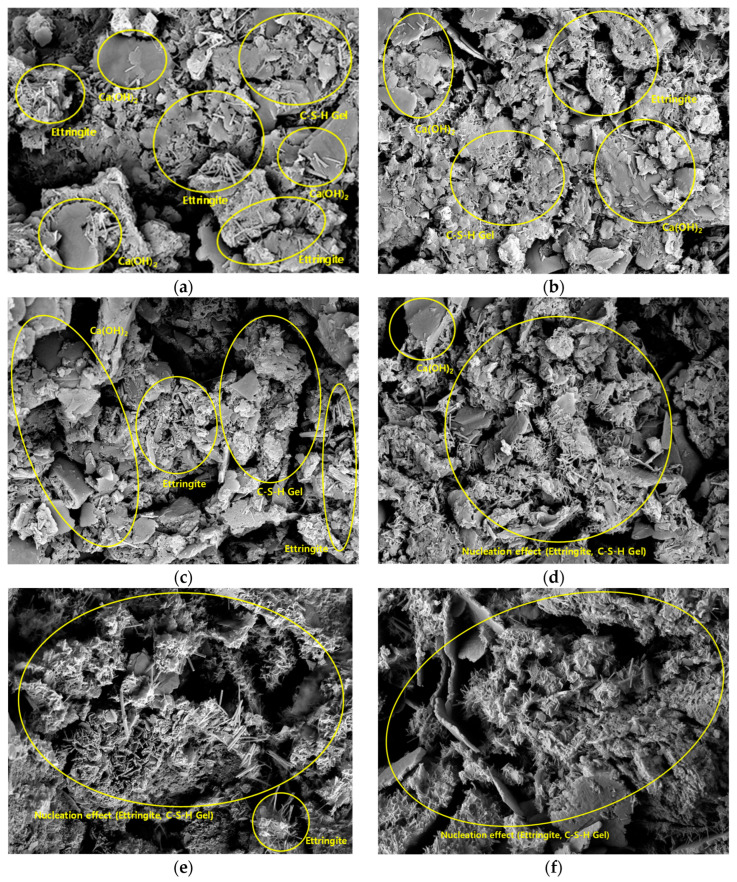
28-d SEM images for different W/B values (×10,000). (**a**) W/B = 500%; (**b**) W/B = 450%; (**c**) W/B = 400%; (**d**) W/B = 350%; (**e**) W/B = 300%; (**f**) W/B = 250%.

**Figure 8 materials-16-01511-f008:**
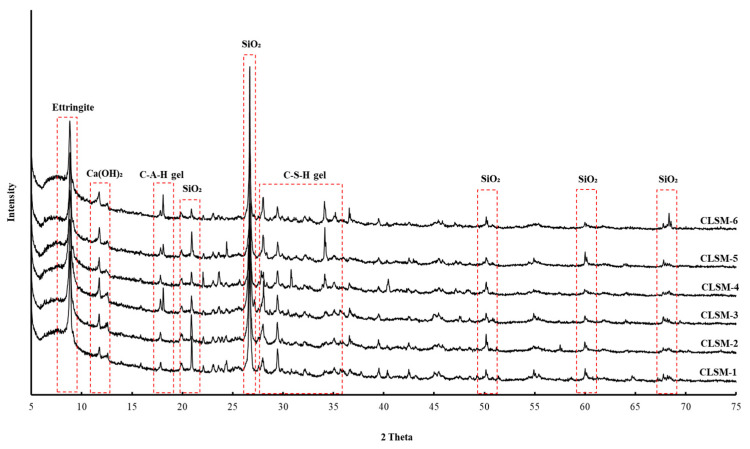
7-d XRD results for different W/B values.

**Figure 9 materials-16-01511-f009:**
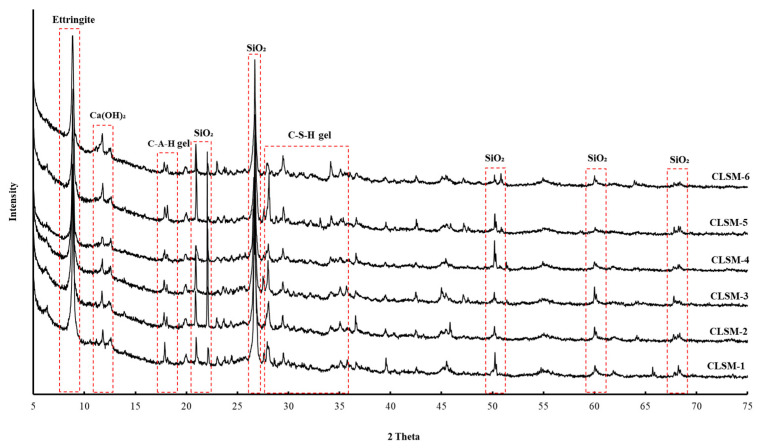
28-d XRD results for different W/B values.

**Figure 10 materials-16-01511-f010:**
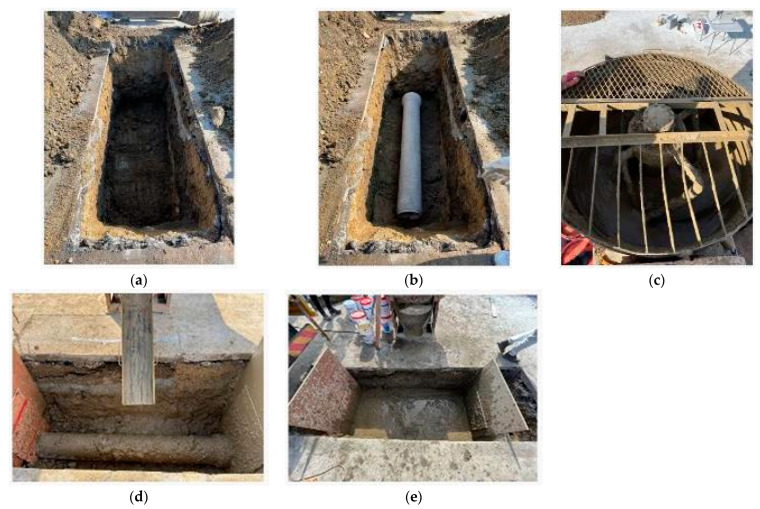
Photographs of the field applicability evaluation of the CLSM using stone sludge. (**a**) excavation; (**b**) pipe landfill; (**c**) CLSM mixing; (**d**) pouring; (**e**) construction completed.

**Figure 11 materials-16-01511-f011:**
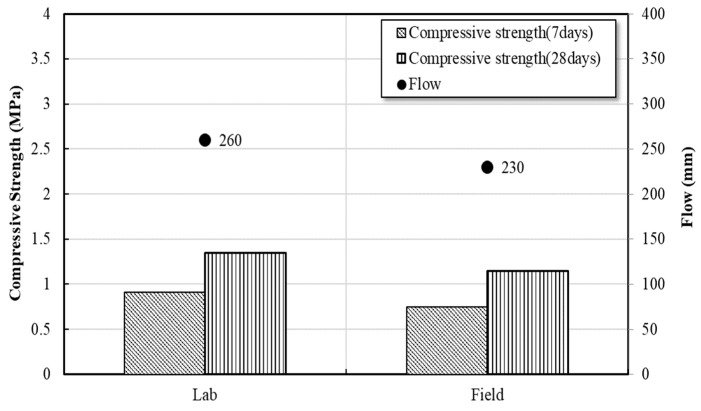
Field and lab test evaluation results for the CLSM using stone sludge.

**Figure 12 materials-16-01511-f012:**
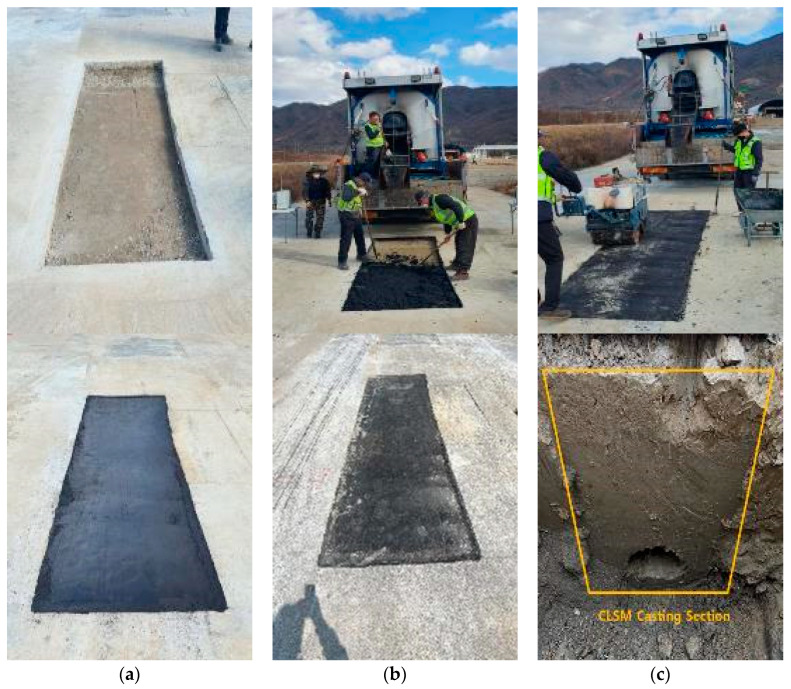
Field application monitoring results for CLSM using stone sludge. (**a**) After construction; (**b**) After 5 months; (**c**) CLSM cross-section.

**Table 1 materials-16-01511-t001:** Evaluation results for the basic properties of the stone sludge.

Property	Unit	Results
CBR test (correction CBR)	%	13.6
Maximum dry density	g/cm^3^	1.694
Liquid limit	%	39.36
Plastic limit	%	20.13
Plastic index	-	19.23

**Table 2 materials-16-01511-t002:** Water-content test results for the stone sludge.

Division	Sample-1	Sample-2	Sample-3
Wet soil (g)	902	832	790
Dry soil (g)	645	594	568
Water (g)	257	238	222
Water content (%)	39.85	40.07	39.09
Average water content (%)	39.67		

**Table 3 materials-16-01511-t003:** Physical properties and chemical composition of the cement.

Density (g/cm^3^)	Fineness (cm^2^/g)	Chemical Composition (%)						
		SiO_2_	Al_2_O_3_	Fe_2_O_3_	CaO	MgO	SO_3_	Ig.loss
3.14	3492	21.1	4.64	3.14	62.8	2.81	2.13	2.18

**Table 4 materials-16-01511-t004:** Mix proportion of CLSM using stone sludge (kg/m^3^).

Test ID.	Cement (kg)	Water (kg)	Inorganic Sludge (kg)	Amount of Water in Sludge (kg)	W/B
CLSM-1	148	510	797	232	500%
CLSM-2	164	508	792	230	450%
CLSM-3	183	504	788	229	400%
CLSM-4	208	500	782	227	350%
CLSM-5	240	495	773	225	300%
CLSM-6	283	487	762	222	250%

**Table 5 materials-16-01511-t005:** Hardness test results for the CLSM using stone sludge.

No.	Soil Penetrometer (mm)			
	1 h	2 h	3 h	4 h
1	1.5	3.1	3.8	4.5
2	1.8	3.0	3.9	4.3
3	1.6	2.8	3.7	4.7
4	1.9	3.2	4.1	5.0
Average	1.70	3.03	3.88	4.63

## Data Availability

Data sharing is not applicable.

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
