# Peer review of "Microstructure Analysis and Mechanical Properties of Backfill Material Using Stone Sludge"

_materials, 2023, doi:10.3390/ma16041511_

Round 1

Reviewer 1 Report

Comments on the under-reviewing paper – Materials Journal

The paper under review states that increases the utilization of stone sludge, which replaces only part of the existing concrete materials, to conserve natural resources and expand the recycling of industrial byproducts.

Comments:

1- The literature review needs to be intensely enriched and narrowed with studies that utilized stone sludge in backfill applications – more emphasis on what distinguishes the paper from other work mentioned in the literature review.   

2- Line 81, why is maximum dry density presented?  Discuss the variability of the densities and number of samples being tested – please consider this comment for other parameters as well.  

3- Line 82, no reference for given standard – make sure all standards are referenced in the paper.

4- Line 91, is the company names A or an arbitrary number?

5- Line 94, Section 2.2 mixing design, on what bases optimal mixing is chosen – not clear?

6- Line 95, Section 2.2 mixing design, on what bases optimal mixing is chosen – not clear?

7- Line 96, Section 2.2 mixing design, on what bases optimal mixing is chosen – not clear?

8- Is it the water-cement ratio or there is another binding?

9- Lin 96, Section 2.2 mixing design, the author mentioned “by selecting the main experimental 95 factors such as the stone sludge characteristic analysis”…needed to be rewritten in detail how characteristic analysis is one parameter/factor. This entire paragraph needed to be restructured.

10- In line 97, the Authors mentioned the stone sludge-to-binder ratio (S/B), but this value is not mentioned in table 4, illustrating only the water-to-binder percentage (W/B).

11- Table 4, on what bases are these mixing ratios are chosen? What was the basis for selecting these proportions of the stone sludge in the investigated mixtures?

12- In the mixing design section 2.2, if the authors prepared a different mixture to be considered as a control mix (without sludge), that will be preferred for comparison purposes and directly help interpret XRD results.

13- Line 101, Section 2.3.1, how mixing design evaluated – discuss in the test?

14- The author mentioned ..”to meet each standard”.. But no specific standards are discussed for each parameter.  This entire section needed to be restructured.  

15- Line 108, Section 2.3.2, - please reference standards mentioned the text and the entire paper.

16- In the experimental section, it is good to have photos from equipment, samples, mold, etc. used in the experiment if applicable.

17- Line 127, Section 2.3.4, please proofread the English language works– e.g. height instead of tall, re-excavation, etc.  Some words may confuse the reader.

18- In line 162, what does ACI Committee mean? No reference provided?

19- In section 2.3.7, many phrases are not discussed in the prior section, for example – the follow-up process after construction, what is the mean? ….. confirmed through a characteristic analysis, again the reader may get confused about what the authors want to discuss.  Please check.

20- Line 160, what do the authors mean by quality standards – what are they? May need a brief explanation of why ACI Committee Report 229 was selected among other standards.

21- All standards used in the text should be referenced in text

22- In Figures 4 and 5, How did you distinguish between ettringite, C-A-H, C-S-H, and CO(OH)2 from SEM images?  Please discuss in detail the features in the image in 7 or 28 days that are associate with each one of these.

23- Line 255, section 3.6, Derivation of optimal mixing ratio!  Discuss the optimization method used as I believe it is rather a selection of mixes that satisfy standard criteria than a formal optimization technique. 

24- Section 2.3.4 (Compressive strength test), authors conducted this test according to ASTM D 4832, but it is not considered in the references list!!

25- Line 221, What is the role of Portlandite Ca(OH)2, and why is a large amount of this mineral associated with a high W/B ratio?

Author Response

I extend my sincere gratitude to the reviewers for their detailed and comprehensive feedback. I have carefully considered their suggestions and comments in revising this paper, and I am confident that the revisions have greatly improved the quality of the work. I hope that this revised version will meet their expectations.

Reviewer 2 Report

1. Picture problem:

(1) In Figure 4, the marking form of (c) and (f) shall be consistent with the other four groups.

(2) In Figure 6-7, the font of the text in the picture is too small and blurry, so the image resolution and text size should be adjusted. Make sure the text size in the full text image is the same.

2. It only stays in the description of the experimental process and the result statement, which is deficient in theoretical principles and research significance.

3. When carrying out the mechanical compression test, only the mechanical strength of 7 days and 28 days was measured, and the mechanical strength of 14 days was ignored. For the concrete compression test, the 14 day mechanical strength is also an important data, which is recommended to be supplemented.

4. As the research of the test is insufficient, the test results should be further analyzed.

5. For practical engineering applications, only 5 months of on-site monitoring will determine that it can be used. Is the time too short? Or provide relevant specifications to demonstrate that the monitoring time of the author meets the requirements.

6. Is it not rigorous enough to replace data detection by visual observation of the fallibility and broken pipe around the pipeline?

7. For the difference between field test and laboratory results, the author needs to consider more reasons besides the weather and mixer mentioned in the manuscript.

Author Response

(The authors gave the same response as above.)

Reviewer 3 Report

The authors have presented a study on use of recycled sludge in CLSM, an area of potential interest to readers. Technical and editorial improvements are needed to derive more useful information from the paper.
Significant editorial work is needed for English language. 

Technical comments:

Abstract: more info is needed on the test methods performed. When the authors state the material "met the standards", what standards are being references?

Introduction: Several prior studies are listed-but what were the results of these studies? Also, the differences between the previous studies and this one need to be more clearly outlined.

Materials and Methods:
Section 2.2: Avoid the use of the first person ( I, we) 
Table 1 (mislabeled as 4): The lack of a control CLSM without sludge is a large oversight, as it makes comparisons more difficult. Also, how many samples of each mix were made? How many tests were performed on each mix for strength, fluidity, bleeding, etc?
Also, it looks like the amount of sludge was not varied-any reason for this?
Section 2.3: Why would SEM and XRD be used to determine optimal mixing ratio? Plastic and hardened properties would be a far more useful comparison.
2.3.4 More information is needed on the re-excavation. How was it done, and what were the conditions during the period underground?
2.3.7: 5 months seems like a short field time, considering subsidence and other issues can take years.

Results:
3 (all sections): How many tests were performed on each mix?
3.2: Interesting that w/b seemed to have no effect on bleeding. Do the authors have any explanation for this behavior?
3.7 The authors state the compressive strength increased by 10% compared to the lab, but it appears to have decreased.

Author Response

(The authors gave the same response as above.)

Round 2

Reviewer 1 Report

Thank you for including my suggestions to improve the manuscript. 

Author Response

Thank you

Reviewer 2 Report

Looking forward to your more scientific research results.

Author Response

Thank you